# Audiophonologopedic Telerehabilitation: Advantages and Disadvantages from User Perspectives

**DOI:** 10.3390/children11091073

**Published:** 2024-08-31

**Authors:** Maria Lauriello, Anna Maria Angelone, Sara Iannotti, Eleonora Nardecchia, Benedetta Scopano, Alessandra Fioretti, Irene Ciancarelli, Alberto Eibenstein

**Affiliations:** 1Department of Biotechnological and Applied Clinical Sciences, University of L’Aquila, 67100 L’Aquila, Italy; maria.lauriello@univaq.it (M.L.); alberto.eibenstein@univaq.it (A.E.); 2Department of Life, Public Health and Environmental Sciences, University of L’Aquila, 67100 L’Aquila, Italy; annamaria.angelone@univaq.it (A.M.A.); irene.ciancarelli@univaq.it (I.C.); 3Centro di Audiofonologopedia, 00199 Roma, Italy; sara.iannotti12@gmail.com; 4Speech and Language Therapy, Department of Life, Public Health and Environmental Sciences, University of L’aquila, 67100 L’Aquila, Italy; eleonora.nardecchia@gmail.com (E.N.); benedettascopano997@gmail.com (B.S.); 5European Hospital, 00149 Rome, Italy

**Keywords:** e-health, telemedicine, telerehabilitation, COVID-19, neurodevelopmental disorders, hearing loss, user perspective

## Abstract

Introduction: Technological advancements and the COVID-19 pandemic have highlighted the importance of digital tools for patient care and rehabilitation. This study explores user perspectives on telerehabilitation, comparing it to traditional methods and identifying criteria for determining its suitability for different patients and clinical conditions. Methods: This study was carried out during the period of May–September 2021. Questionnaires were administered to 48 users in rehabilitation for audiophonologopedic and neurodevelopmental disorders in three rehabilitation centres in central Italy. Results: The user responses predominantly emphasize the benefits of time saving (68.75%) and cost-efficiency (37.5%), specifically regarding time saving due to travel and expenses incurred to go to where the therapy is carried out. The disadvantages include increased distraction (60.42%) in following the instructions remotely and logistic problems (39.58%). Patients with hearing loss were subjected to a larger number of telerehabilitation sessions, positively rating this alternative method. Patients with speech and language delay and autism spectrum disorder (ASD) prefer traditional treatment. Discussion: This study reveals a favourable perception of telerehabilitation as a therapy approach to be regarded as a supplement or temporary option to the irreplaceable face-to-face one. More research, as well as a larger sample sizes, will be useful to increase the significance of the correlations reported in this study.

## 1. Introduction

Telemedicine, now defined as telehealth or e-health [1], is becoming a popular alternative method to provide healthcare services supported by Information and Communication Technologies (ICTs). It is especially useful when a direct patient–healthcare provider relationship is not possible [2,3], with the aim of ensuring hospital–territory continuity, care accessibility, and cost savings, while maintaining service quality and universality [4]. The large number of papers and research works on telemedicine highlight an increasing interest in this field [5], especially after its wide use during the COVID-19 pandemic, when services were limited by lockdown measures imposed by Governments to contain the spread of SARS-CoV-2 infection. Due to the lockdown, patients experienced delays, discontinuances, and cancellations to their treatments, weakening their effectiveness [6]. As a result, the pandemic created the need for new health services by enhancing remote performances and developing national guidelines for telemedicine and telerehabilitation [7,8]. Rehabilitation services found new opportunities through digitisation, which led to the telerehabilitation, a specialized branch of telemedicine for rehabilitation, consultation, and monitoring activities. Telerehabilitation has different applications, including motor and cognitive rehabilitation, as well as occupational, communication, deglutition, behavioural, cardiological, and respiratory disorders. Furthermore, telerehabilitation can be used for the prescription and evaluation of aids, orthoses, and prostheses. Telerehabilitation required technological supports (tablet, smartphone, computer), as well as a stable and high-quality Internet connection and a therapy-dedicated station. The sudden interruption of traditional rehabilitation approaches during the pandemic particularly hit those patients with chronic diseases, rare diseases, neurodevelopmental and psychiatric disorders, psychophysical disabilities, and patients who need medium- and long-term continuity of care [8]. The guidelines provided by the Italian Ministry of Health [8] were characterized by measures implemented towards mental health protection and vulnerable subjects, particularly pediatric ones [9,10,11]. The purpose of this medical and political act goes far beyond the reduction in the consequences of a specific disorder, but aims to develop the adaptive function of the subject, to realize self-sufficiency, and to improve the quality of life, sociality, and work contribution [12]. Therefore, any problem in accessing a rehabilitation treatment means irreversibly jeopardizing psychophysical development, defining an irreversible disability and the failure to adapt in society with all the related consequences [13]. Because of its multidisciplinary characteristic, telerehabilitation is particularly well-suited for the delivery of the audiophonologopedic therapy [14,15,16,17,18,19,20,21,22,23,24,25]. The chance of a “tailored” therapy based on the patient’s needs and preferences, combined with the flexibility of a remote therapeutic approach, contributes to the effectiveness of the rehabilitation process [26,27]. Guidelines recommend the evaluation of the available resources related to the territory, family, and social environment, as well as the potential obstacles to the treatment, in order to adjust the therapeutic plans to empower the appropriateness of the intervention with the goal to achieve positive outcomes. The evaluation of functional and care outcomes is carried out to monitor the rehabilitation progress together with the user and/or caregiver’s satisfaction and treatment perception (e.g., Patient Reported Outcome Measure) [8]. Few data are currently available on the satisfaction perceived by telerehabilitation actors (parents, users, clinicians, and therapists). Scientific studies are available in the literature that highlight the inconsistent outcomes of telerehabilitation, generating scepticism also among parents regarding the use of a remote assistance support [28,29]. The aim of this study on audiophonologopedic telerehabilitation is to collect scientific data regarding the experience reported by the users of this tool and which benefits and drawbacks were emphasized [30]. Based on the results of this study, together with those provided by further research, we can better outline specific fields of intervention with telerehabilitation: when it is recommended, when it is not recommended, and even when it can give rise to negative outcomes, always considering the clinical condition and the age of the patient, as well as the available resources. So far, the therapeutic options developed with the telerehabilitation have proven to be feasible both from an organizational and economic point of view, but only a prolonged experience over time supported by scientific evidence will confirm the full clinical validity of telerehabilitation and its actual incorporation into clinical practise as a supplement to the traditional method.

## 2. Materials and Methods

The reference population of this study, conducted from May to September 2021, includes patients treated in three rehabilitation centres in central Italy. These are the patients whose parents have accepted to participate in this study and to fill out the scheduled questionnaire. The heterogeneity of the population included in the study is due to the inclusion criteria: no limits for age and type of pathology were set with the result of eight different pathological frameworks. We did not use strict exclusion criteria because we collected preliminary data to evaluate as many patients as possible, as there are not many similar studies already performed using user perspectives. All the patients carried out speech therapy aimed at the re-education or education of expressive, receptive, and communication language. The subjective questionnaire was provided to the users who received remote speech rehabilitation during the COVID-19 pandemic. The users were required to respond anonymously, and, in the case of minors, young children, or patients with a clinical condition that prevented a complete comprehension, the questionnaire was completed in the presence of a parent or caregiver, who eventually reported their answers. Participation was on a voluntary basis, and every user signed the informed consent form to the processing of anonymous data for the research project, approved by the Internal Review Board (Prot. N. 41818, 5 April 2022, University of L’Aquila). A questionnaire of 17 questions was developed, which provides general user information (age, origin, and pathology), investigates rehabilitation delivery methods (synchronous or asynchronous, direct, or indirect) and the type of platforms used, highlights the user preference (face-to-face or remote therapy), and assesses the perception of improvements achieved after telerehabilitation (Appendix A). After a brief explanation of the study and research objectives, the questionnaire and informed consent were supplied in paper form; users completed both and placed the two papers into separate boxes. Some participants electronically signed the consents and completed the survey providing responses through email or WhatsApp/Telegram. Others answered the structured interview-style questionnaire in the presence of the speech therapist and parent/caregiver. After data collection, Stata 15 software(version:15.1) was used (Statistics/Data Analysis TA USA, Stata Corp LP, College Station, TX, USA). Stratification by pathology, treatment method, improvements, distractions, and time saving was carried out to evaluate the distribution of frequency. A χ^2^ test was used to determine whether or not there was a statistically significant difference between the groups. A *p* value ≤ 0.05 was considered significant.

## 3. Results

The reference sample for the analysis consists of 48 users (42 children and 6 adults; mean of users age 11.8 years old; SD 10.7). The pathologies of the participants are reported in Table 1.

It is important to note that all users began rehabilitation treatment prior to the COVID-19 emergency, and none claimed that they would utilize this alternative approach until 2020.

When the reference centre’s multidisciplinary team proposed conducting remote sessions, 76% of users felt reassured about the idea of not interrupting treatment; instead, 19.57% were sceptical about this method.

Telerehabilitation requires the possession of Internet connection and digital devices, such as computers, tablets, and mobile phones: only a small group of patients (10.42%) did not have the essential tools to access to remote rehabilitation. We did not evidence a significant difference in the results based on the type of device used for telerehabilitation because all the users utilized computers. The platforms, dedicated web sites, used and the messaging and video calling apps used are reported in Table 2.

Another fundamental element to consider for the success of telerehabilitation is the autonomy of the user during the session. Few patients did not need to seek assistance (12.50%) during the session. The majority of patients needed help from a parent/caregiver in every session (37.50%) or only the first time (39.58%). In other cases, a speech therapist guided the user in case of need (10.42%), asking for the help of the parents by phone or during the session of telerehabilitation.

Speech therapy involved individual sessions (72.34%) or mixed ones, also including co-presence sessions (27.66%). A co-presence session was composed by more users together in a session, where everyone had their own speech therapist.

Regarding the number of weekly telerehabilitation sessions, the users mainly participated once a week (34.04%) and twice a week (55.20%).

The importance of therapy continuity is particularly underscored in replies relating to session frequency: more than 67% of patients maintained the same involvement, while 20% reported increased attendance with telerehabilitation.

Although telerehabilitation has been expressly requested by patients in the later phases of the COVID emergency, it is essential to point out that the majority of users believe that face-to-face treatment is more helpful for the management of various pathologies [Table 3].

Subjects with an intellectual disability or ASD did not perceive improvements via telerehabilitation, which was seen as more of a maintenance of the obtained results. Patients with hearing loss, instead, have described perceived improvements, and some have even reported better results with remote treatment [Table 4].

Those who filled out the questionnaire were asked to express the advantages they perceived with telerehabilitation. The main advantages are time saving (68.75%), cost-efficiency (37.5%), and increased concentration (20.83%) [Figure 1].

The first two are related to a more organizational–management aspect of rehabilitation, which reflects the caregiver’s point of view: remote treatment allows you to avoid long journeys to reach the rehabilitation centre, a non-secondary factor in cities like Rome, and allows you to save on the cost of fuel, tolls, and transportation tickets.

Statistically, significant associations have been identified to confirm this: almost all patients with hearing loss (92.86%; *p*-value = 0.020), those who prefer telerehabilitation (100%; *p*-value = 0.041), and also users who rather use the traditional method (59.46%, *p*-value = 0.041) consider time saving an advantage. In general, all the patients interviewed recognize that time saving in remote therapy is essential, although this is not considered by all to be the best way of providing treatment for their pathology (*p* = 0.039). Furthermore, 72.97% of patients claim to be distracted during face-to-face therapy and 80.00% of them claim to maintain concentration with remote therapy (*p* = 0.005) (Table 5)

Logistical difficulties (39.58%) and more distraction (60.42%) are, according to the users, the main telerehabilitation disadvantages [Figure 2].

More distraction has a statistically significant association with ASD (88.89%; *p*-value = 0.05) and intellectual disability (100%; *p*-value = 0.020). So, based on these statistical data, telerehabilitation is not particularly appropriate for these disorders. In total, 64.29% of patients with hearing loss report an absence of distractions.

There is a statistically significant association (*p* = 0.033) between distraction and pathologies. However, this association is not statistically significant (*p* = 0.112) between time saving and pathologies (Table 6).

The analysis shows a significant association between the distraction variable and three individual pathologies: intellectual disability (*p* = 0.033), ASD (*p* = 0.05), and hearing loss (*p* = 0.025) (Table 7).

## 4. Discussion

The technological progress, together with the large number of patients increasingly accountable for their own health, with an active part in the decision-making process for the assessment of the therapeutic programme, allowed for a quick upward evolution in the use of digital technology for rehabilitation. Furthermore, the need to guarantee an adequate level of assistance during the pandemic forced the use of telerehabilitation in clinical practise, both in Italy and abroad. This surge was also aided by the progressive interest in the scientific literature on this topic and the effectiveness of telemedicine therapies and telerehabilitation [31,32]. However, these studies all have the limitation of research being conducted with a specific focus on the qualitative values and not on the quantitative ones, resulting in a loss of statistical strength. This study aims to collect and examine the advantages and disadvantages of telerehabilitation from the user’s point of view, a topic that has received little attention in the scientific literature.

The same Italian guidelines on remote rehabilitation consider the patient’s opinion to be critical to better evaluate the indications for an effective therapy conducted in a remote mode with improved multidisciplinary management.

The sample consists of patients from rehabilitation centres in central Italy: 48 users, or their parents/caregivers, who completed the questionnaire. The patients were treated for hearing loss, ASD, intellectual disability, speech and language delay, specific learning disorders, ADHD, and cleft palate.

All participants started with face-to-face treatment before the pandemic, and they had never used telerehabilitation before, despite the fact that it was already available. Since the implementation of the lockdown measures, the use of a remote mode therapy has been recommended to prevent interruption to or discontinuity in rehabilitation.

Telerehabilitation was supported by nearly three-quarters of the patients who were concerned about the reduced frequency or interruption of the therapy. However, in a small percentage of cases, doubts and worries were expressed about this new mode of intervention. Fortunately, almost every family already had the technological support (tablet, smartphone, computer) for telerehabilitation, as well as a stable and high-quality Internet connection and a therapy-dedicated station. Despite the rapid digitalisation of society, some interviewees said that they did not have the appropriate devices at first and thus had to obtain them quickly. This highlights the presence of population groups that do not have easy access to technological progress due to economic constraints.

After the first emergency phase (May 2020), not all the patients returned to their usual home, and some of them wanted to continue the telerehabilitation for their own needs or, always in agreement with the therapist, opted for a combination of the two modalities from remote and face-to-face treatment. The adhesion to the therapy sessions was stable for many users, decreased only for a small number of patients, and increased for nearly a fifth of the interviewees. This can be explained by the better organizational ease with which the remote modality was used, as it was also used for work and educational purposes during the pandemic.

We attempted to establish the relationship between the modality of the therapeutic sessions and the disease, but the current amount of data does not allow us to consider them as significant (*p*-value > 0.05). However, we can find some correlations showing that individual sessions are carried out mostly by patients with speech and language delay and hearing loss. Patients with hypoacusis have performed more sessions of both types, individual and co-presence.

The collaboration between the professionals and the families involved in the treatment was essential, especially during the early phases of telerehabilitation requiring some training and adaptation. Many users stated that they needed assistance and were initially guided by a speech therapist.

Another key component to examine to evaluate telerehabilitation is the perception of specific improvements. More than half of the interviewees believe that the progress achieved was similar in both modalities, and many of them think that remote rehabilitation can be as useful as valuable for maintenance and to consolidate the results obtained with the therapy.

With the limitations of the current data, it should be considered that despite the positive qualities noted with telerehabilitation, face-to-face rehabilitation is seen as the most effective method, satisfying the preferences of patients with ASD and speech and language delay. For the patients with hearing impairment, remote rehabilitation can be an instrument that allows them to achieve similar results and sometimes results superior to those obtained in person. Although statistical significance for some correlations should be studied with a larger sample size, we can still consider positively the results obtained during telerehabilitation, as well as the fact that was not to interrupt a non-delayable treatment. The most obvious benefits are time saving and economic convenience, related to the organizational aspect of rehabilitation. The understanding of these advantages must be linked to the fact that the majority of the interviewed users were from Rome and its suburbs, and with telerehabilitation they would save time otherwise wasted in traffic and travelling large distances. It is also important to consider the economic benefit for a reduced use of cars, as well as the use of public transportation and the cost of the ticket. There is a statistically significant correlation between time saving and hearing loss pathology. Furthermore, this advantage is recognized by both individuals who believe that telerehabilitation is the most effective treatment for their condition and those who prefer traditional treatment. A minority of users and caregivers, observing patients’ behaviour, noticed an increased concentration with the remote mode. There is a significant correlation for this advantage between those who did not see the increase in concentration and the users who saw telerehabilitation as a maintenance treatment only.

This allows us to speculate about reported improvements and a lack of attention, which is supported by another statistically significant correlation between observed improvements and one of the perceived disadvantages, more distraction during telerehabilitation sessions. As a result, not seeing improvements can be attributed to the distraction caused by distance and the digital barriers. This disadvantage is also associated with pathologies such as intellectual disability and ASD, both of which are difficult to handle by the therapist remotely. Other drawbacks identified by users include logistical issues related to the stability of Internet connection and the availability of a station dedicated to therapy, difficulty in understanding the exercises to be performed, and a lack of the social aspect that is normally guaranteed by attendance at centres. The emotional side was partially compensated by the ability to establish eye contact via digital means. Less easy, however, was the total home management of patients with complex clinical situations, in which the caregiver often received little assistance [33,34]. Verma et al. evaluated 27 unilateral pediatric cochlear implantees in the age range of 2–11 years, divided into two groups based on the therapy modality, telerehabilitation and face-to-face. The study indicated that the conventional method of the speech–language and auditory rehabilitation was far better compared to the telerehabilitation services with the WhatsApp video calling platform, as majority of the population were from lower socioeconomic backgrounds with poor educational backgrounds. The limitations reported with the teletherapy mode were connectivity issues, poor sound quality, and poor visibility [35].

### 4.1. Limitations

The small sample size of this study may affect the representativeness and statistical significance of the results. In future research, we will increase the sample size to improve the reliability of the results. The current research mainly focuses on patients with hearing loss, language delay, and autism spectrum disorder. We will include more types of rehabilitation patients in future research to obtain more comprehensive conclusions. We will also introduce future studies in which different age groups are separated, considering the subjectivity of the proposed questionnaires. It would also be very interesting to compare the results obtained by groups of patients of the same age and with the same pathology, who performed the rehabilitation course in person versus those who performed part of it in telerehabilitation.

### 4.2. Follow Up

The use of telerehabilitation did not stop in the Audiophonologopedic Center of Rome with the end of the pandemic emergency, but it continued. Currently, three users are being followed with exclusive telerehabilitation, including two girls (15.8 and 14.1 years old) with sensorineural hearing loss, wearing cochlear implants, and a child (7.11 years old) suffering from a neoplastic disease being treated at a hospital and, therefore, in a frail condition not compatible with in-person therapy. The reasons, in the case of the two girls, are to be ascribed to the distance of their homes from the centre’s headquarters and the economic difficulties of both families that mean it would not be possible to accompany them, losing hours of work and forcing them to spend a greater amount of energy taken away from studying and socialization with peers. For the remaining 193 users affected by the types of pathologies illustrated in the materials and methods, telerehabilitation is used in a systematic and integrated way in the service delivery processes, with the exception of children with autism spectrum disorder and/or a chronological age of under 6 years, as in the following cases:-To encourage the recovery of therapies not provided, even at times not foreseen by the rehabilitation project;-To allow them to benefit from the treatment at the scheduled time in the event of an obstacle to attendance at the centre due to unexpected family reasons;-To guarantee the treatment in case of mild ailments/convalescence phases, which do not allow attendance in person, but are compatible with rehabilitation work.

In all cases, the telerehabilitation option provided greater flexibility and guaranteed the continuity of work, reducing the number of absences with both organizational advantages for the structure and clinical–rehabilitative ones for the users. A new questionnaire plans to record the advantages and disadvantages of the hybrid use of telerehabilitation.

A study about telerehabilitation in families with children with neurodevelopmental disorders during the lockdown caused by the COVID-19 showed high levels of participation and satisfaction. The authors suggested sharing the resources for their applicability in other countries where families have similar needs conditioned by COVID-19 [36].

## 5. Conclusions

This study is hampered in part by the small number of participants, yet this does not prevent us from finding interesting observations.

Remote rehabilitation has undoubtedly played a functional role in the evaluated experience, ensuring treatment continuity and avoiding the unsuitable psychophysical development of the children. Further studies may assess its actual future relevance as an integration method in the rehabilitation path or as an alternative in case of need, especially as this field of study represents a research stimulus for the clinical, rehabilitation, and technological sectors.

## Figures and Tables

**Figure 1 children-11-01073-f001:**
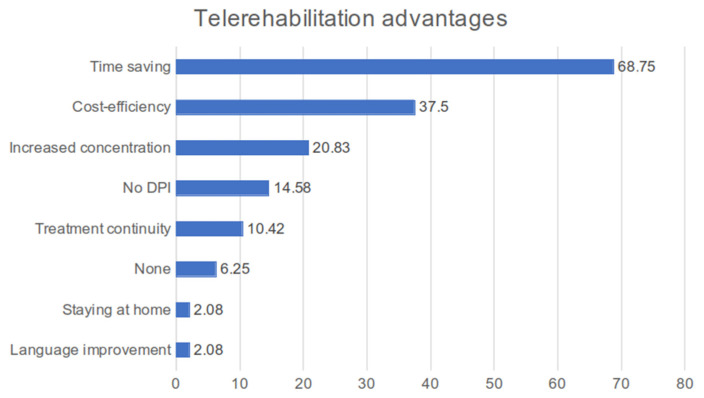
Histogram of answer percentages about telerehabilitation advantages according to users’ points of view. DPI: individual protection devices.

**Figure 2 children-11-01073-f002:**
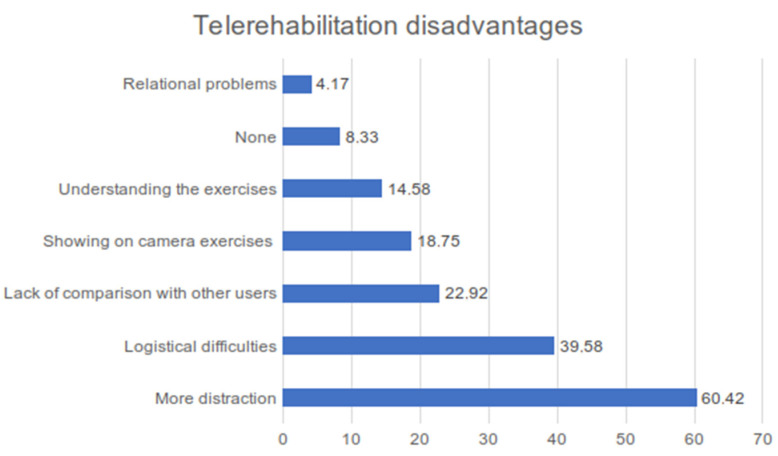
Histogram of answer percentages about telerehabilitation disadvantages according to users’ points of view.

**Table 1 children-11-01073-t001:** Pathologies reported by the participants. ASD: autism spectrum disorder; ADHD: attention deficit hyperactivity disorder.

Pathologies	%
hearing loss	29.17%
ASD	18.75%
speech and language delay	16.67%
specific learning disorder	14.8%
intellectual disability	14.58%
ADHD	4.17%
cleft palate	2.08%

**Table 2 children-11-01073-t002:** Platforms, dedicated web sites, and messaging and video calling apps used by the participants.

	%
trainingcognitivo.it (accessed on 24 August 2024).	8.70
Both Skype (iOS 8.123.0.203) and Zoom (6.1.11.45504)	6.52
Only Skype	6.52
Both trainingcognitivo.it and wordwall.net (accessed on 24 August 2024)	4.35
Only wordwall.net (accessed on 24 August 2024).	2.17
WhatsApp (Version 22.13.74)	2.17
Not specified	19.57

**Table 3 children-11-01073-t003:** Users’ response percentages related to the pathology and treatment methods. ASD: autism spectrum disorder; ADHD: attention deficit hyperactivity disorder.

KERRYPNX	Treatment Method Which is Perceived as the Best						
▪ Pathology	Telerehabilitation		In Person		Both		
▪ n Total	n	(%)	n	(%)	n	(%)	*p* Value
▪ Speech and language delay 8	0	(0.00)	8	(100.00)	0	(0.00)	*0.492*
▪ Specific learning disorder 7	1	(14.29)	6	(85.71)	0	(0.00)	
▪ Intellectual disability 7	1	(14.29)	5	(71.43)	1	(14.29)	
▪ ASD 9	0	(0.00)	8	(88.89)	1	(11.11)	
▪ Hearing loss 14	3	(21.43)	7	(50.00)	4	(28.57)	
▪ ADHD 2	0	(0.00)	2	(100.00)	0	(0.00)	
▪ Cleft palate 1	0	(0.00)	1	(100.00)	0	(0.00)	

**Table 4 children-11-01073-t004:** Users’ response percentages related to the pathology and noted improvements. ASD: autism spectrum disorder; ADHD: attention deficit hyperactivity disorder.

Have You Perceived Any Improvements?							
	No, I Have Not. It Was Useful as Maintenance		Yes, I Have Perceived the Same Grade of Improvements		Yes, I Have Perceived Better Improvements with Telerehabilitation		
▪ Pathology	n	(%)	n	(%)	n	(%)	*p* Value
▪ Speech and language delay	2	(25.00)	5	(62.50)	1	(12.50)	*0.246*
▪ Specific learning disorder	4	(57.14)	3	(42.86)	0	(0.00)	
▪ Intellectual disability	5	(71.43)	2	(28.57)	0	(0.00)	
▪ ASD	5	(62.50)	2	(25.00)	1	(12.50)	
▪ Hearing loss	1	(7.14)	11	(78.57)	2	(14.29)	
▪ ADHD	1	(50.00)	1	(50.00)	0	(0.00)	
▪ Cleft palate	0	(0.00)	1	(100.00)	0	(0.00)	

**Table 5 children-11-01073-t005:** Percentages of time saving and distraction reported by patients with telerehabilitation, in person and both.

Treatment Perceived as the Best	Time Saving		No Time Saving		*p* Value	Distraction		No Distraction		*p* Value
	n	%	n	%		n	%	n	%	
Telerehabilitation	5	100.00	0	0.00	*0.039*	1	20.00	4	80.00	*0.005*
In person	22	59.46	15	40.54		27	72.97	10	27.03	
Both	6	100.00	0	0.00		1	16.67	5	83.33	

**Table 6 children-11-01073-t006:** Percentages of time saving and distraction reported by patients in telerehabilitation.

Pathology	Distraction n (%)		No Distraction n (%)		*p* Value	Time Saving n (%)		No Time Saving n (%)		*p* Value
Speech and language delay	5	62.50	3	37.50	*0.033*	3	37.50	5	62.50	*0.112*
Specific learning disorder	3	42.86	4	57.14		4	57.14	3	42.86	
Intellectual disability	7	100.00	0	0.00		1	14.29	6	85.71	
ASD	8	88.89	1	11.11		4	44.44	5	55.56	
Hearing loss	5	35.71	9	64.29		13	92.86	1	7.14	
ADHD	1	50	1	50		1	50.00	1	50.00	
Cleft palate	0	0.0	1	100.00		1	100.00	0	0.00	

**Table 7 children-11-01073-t007:** Significant association between the distraction variable and intellectual disability, ASD, and hearing loss.

Pathology	Distraction n (%)		No Distraction n (%)		*p* Value
Speech and language delay	5	62.50	3	37.50	*0.895*
Specific learning disorder	3	42.86	4	57.14	*0.304*
Intellectual disability	7	100.00	0	0.00	*0.033*
ASD	8	88.89	1	11.11	*0.050*
Hearing loss	5	35.71	9	64.29	*0.025*
ADHD	1	50	1	50	*-*
Cleft palate	0	0.0	1	100.00	*-*

## Data Availability

Data are contained within the article or Appendix A.

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
