# Peer review of "Audiophonologopedic Telerehabilitation: Advantages and Disadvantages from User Perspectives"

_children, 2024, doi:10.3390/children11091073_

Round 1

Reviewer 1 Report

Comments and Suggestions for Authors

In this study the Authors reported advantages and disadvantages of audiophonologopedic telerehabilitation from user perspective.

The study is interesting, because it explores user perspectives on telerehabilitation, comparing it to traditional methods and identifying criteria for determining its suitability for different patients and clinical conditions. However, the manuscript would benefit from some revisions:

1.     The introduction section is adequately written, providing all necessary information to the readers.

2.     Materials and Methods: The Author should debate why they made the decision not to set limits for age and disease type: it is difficult to think of being able to equalize the result that comes from a questionnaire taken by a 5-year-old child with that of an adult. Probably the results should be differentiated at least by age group <14 years versus >14 years.

3.     Materials and Methods: Authors should mention the different informed consents that were submitted to children compared to adults.

4.     Results: The Authors might consider evaluating whether there is a difference in results based on the type of device used for telerehabilitation or not and discuss it.

5.     In all the tables it is not clear why some numbers are in bold. The authors should specify this or possibly change it if it is a style error.

6.     Table 3 needs to be modified since the columns and their results appear out of phase with each other.

7.     Results: Regarding the need to seek assistance during the session of telerehabilitation, how it was organized? (line 149)

8.     In the section of Results, the Authors should better explain what a “co-presence session” is (line 154).

9.     Discussion: the Authors should introduce some correlation with the current literature data. In particular, they should explain if their results correlate or not with the literature data in term of distraction variable, time saving, logistical difficulties.

10.  In the Discussion section, the authors should introduce a correlation with current literature data regarding the possibility of telerehabilitation even after the end of the pandemic emergency, as they have included a specific “Follow up” section.

11.  Limitations: Authors should include the need to introduce future studies in which different age groups are separated, considering the subjectivity of the proposed questionnaires. It would also be very interesting to compare the results obtained by groups of patients of the same age and with the same pathology, who performed the rehabilitation course in presence versus those who performed part of it in telerehabilitation.

Author Response

  1. The introduction section is adequately written, providing all necessary information to the readers.Response 1: Thanks for the time spent for the review of the paper

2.     Materials and Methods: The Author should debate why they made the decision not to set limits for age and disease type: it is difficult to think of being able to equalize the result that comes from a questionnaire taken by a 5-year-old child with that of an adult. Probably the results should be differentiated at least by age group <14 years versus >14 years.

Response 2: Thank you for your valuable comment. We have made the decision not to set limits for age and disease type to include more patients with the result of eight different pathological frameworks. Children under 18 years filled in the questionnaire with the aim of the parents. We will differentiate the results by age group <14 years versus > 14 years in a future work increasing the number of patients.

3.     Materials and Methods: Authors should mention the different informed consents that were submitted to children compared to adults.

Response 3:Thank you for your valuable comment. Children and adults filled in the same questionnaire. Children under 18 years filled in the questionnaire with the aim of the parents.

4.     Results: The Authors might consider evaluating whether there is a difference in results based on the type of device used for telerehabilitation or not and discuss it.

Response 4: We deeply appreciate your comment. We added this comment in the results: We didn’t evidence a difference in results based on the type of device used for telerehabilitation because all the users utilized computers.

5.     In all the tables it is not clear why some numbers are in bold. The authors should specify this or possibly change it if it is a style error.

Response 5: Thank you for your valuable comment. We have corrected numbers in bold.

6.     Table 3 needs to be modified since the columns and their results appear out of phase with each other.

Response 6: Thank you for your valuable comment. We have modified the columns as requested.

7.     Results: Regarding the need to seek assistance during the session of telerehabilitation, how it was organized? (line 149)

Response 7: We deeply appreciate your comment. We added this “Few patients did not need to seek assistance (12.50%) during the session. The majority of patients needed help from a parent/caregiver in every session (37.50%) or only the first times (39.58%). In other cases, speech therapist guided the user in case of need (10.42%) asking the help of the parents by phone or during the session of telerehabilitation.

8.     In the section of Results, the Authors should better explain what a “co-presence session” is (line 154).

Response 8: Thank you for your valuable comment. We added this: “A co-presence session is composed by more users together in a session, everyone with the own speech therapist.

9.     Discussion: the Authors should introduce some correlation with the current literature data. In particular, they should explain if their results correlate or not with the literature data in term of distraction variable, time saving, logistical difficulties.

Response 9: We deeply appreciate your comment. We improved the text with these comments: We didn’t find in the current literature data a correlation with our results in term of distraction variable, time saving and logistical difficulties. Verma et al. evaluated 27 unilateral paediatric cochlear implantees in the age range of 2–11 years divided into two groups based on the therapy modality, tele-rehabilitation and face-to-face. The study indicated that conventional method of the speech-language and auditory rehabilitation was far better compared to the tele rehabilitation services with WhatsApp video calling platform as majority of the population were from the lower socioeconomic with poor educational background. The limitations reported with tele-therapy mode were connectivity issues, poor sound quality and poor visibility.

10.  In the Discussion section, the authors should introduce a correlation with current literature data regarding the possibility of telerehabilitation even after the end of the pandemic emergency, as they have included a specific “Follow up” section.

Response 10: We deeply appreciate your comment. We added a comment : “A study about telerehabilitation in families with children with neurodevelopmental disorders during the lockdown caused by the COVID-19 showed high levels of participation and satisfaction. The authors suggested to share the resources for their applicability in other countries whose families have similar needs conditioned by COVID-19.

11.  Limitations: Authors should include the need to introduce future studies in which different age groups are separated, considering the subjectivity of the proposed questionnaires. It would also be very interesting to compare the results obtained by groups of patients of the same age and with the same pathology, who performed the rehabilitation course in presence versus those who performed part of it in telerehabilitation.

Response 11: We deeply appreciate your comment. We added this point in the limitations of the study.

Reviewer 2 Report

Comments and Suggestions for Authors

The topic of telerehabilitation is really very relevant and interesting, especially in pediatric practice. This is due not only to the covid 19 epidemic, but also to the impossibility of transporting a disabled child to a clinic, or the lack of specialized centers in populated areas. In this regard, the development of telemedicine is very important. However, the authors should devote a separate section of the manuscript to describing the progress of the study, it makes sense to add a study scheme (study design). It is also necessary to more clearly display the inclusion and exclusion criteria from the study. The conclusions do not clearly reflect the results obtained by the authors, they devote more space to describing similar studies conducted by other authors earlier. At the same time, a detailed analysis of the results obtained by the authors with a description of further prospects for their use in practical medicine is important.

Author Response

However, the authors should devote a separate section of the manuscript to describing the progress of the study, it makes sense to add a study scheme (study design).

Response 1: Thank you for your valuable comment. We have added your comment in the follow up: “A new questionnaire is planned to record the advantages and disadvantages of hybrid use of telerehabilitation

It is also necessary to more clearly display the inclusion and exclusion criteria from the study.

Response 2: Thank you for your valuable comment. We have reorganized inclusion and exclusion criteria. We did not use strict exclusion criteria because we collected preliminary data to evaluate as many patients as possible as there are not many similar studies already performed using the user perspective.

The conclusions do not clearly reflect the results obtained by the authors, they devote more space to describing similar studies conducted by other authors earlier.

Response 3: We deeply appreciate your comments. Accordingly, we have added the results of other authors.

Verma et al. evaluated 27 unilateral paediatric cochlear implantees in the age range of 2–11 years divided into two groups based on the therapy modality, tele-rehabilitation and face-to-face. The study indicated that conventional method of the speech-language and auditory rehabilitation was far better compared to the tele rehabilitation services with WhatsApp video calling platform as majority of the population were from the lower socioeconomic with poor educational background. The limitations reported with tele-therapy mode were connectivity issues, poor sound quality and poor visibility. [35]

A study about telerehabilitation in families with children with neurodevelopmental disorders during the lockdown caused by the COVID-19 showed high levels of participation and satisfaction. The authors suggested to share the resources for their applicability in other countries whose families have similar needs conditioned by COVID-19. [36]

At the same time, a detailed analysis of the results obtained by the authors with a description of further prospects for their use in practical medicine is important.

Response 4: Thank you for your valuable comment. We reported the comment of the other reviewer

We will also introduce future studies in which different age groups are separated, considering the subjectivity of the proposed questionnaires. It would also be very interesting to compare the results obtained by groups of patients of the same age and with the same pathology, who performed the rehabilitation course in presence versus those who performed part of it in telerehabilitation.